# Cellular Immunity in Obesity: Pathophysiological Insights and the Impact of Bariatric Surgery

**DOI:** 10.3390/ijms26209867

**Published:** 2025-10-10

**Authors:** Tania Rivera-Carranza, Angélica León-Téllez-Girón, Raquel González-Vázquez, Paola Vázquez-Cárdenas, Ana Laura Esquivel-Campos, Felipe Mendoza-Pérez, Martín E. Rojano-Rodríguez, Claudia Mimiaga-Hernández, Juan Carlos Cifuentes-Goches, Omar Edgar Peralta-Valle, Eduardo Zúñiga-León, Rafael Bojalil-Parra

**Affiliations:** 1Colegio de Ciencias y Humanidades, Academia de Nutricion y Salud, Universidad Autonoma de la Ciudad de Mexico, Plantel Casa Libertad, Mexico City 09620, Mexico; claudia.mimiaga@uacm.edu.mx (C.M.-H.); juan.cifuentes@uacm.edu.mx (J.C.C.-G.); 2División de Ciencias Biologicas y de la Salud, Departamento de Atencion a la Salud, Universidad Autonoma Metropolitana, Xochimilco, Mexico City 04960, Mexico; rafaelbojalil@gmail.com; 3Division de Nutriologia Clinica, Hospital General “Dr. Manuel Gea Gonzalez”, Mexico City 14080, Mexico; tellezgiron17@gmail.com; 4Facultad de Medicina, Universidad Nacional Autonoma de Mexico, Mexico City 04510, Mexico; 5Laboratorio de Biotecnologia, Departamento de Sistemas Biologicos, SECIHTI-Universidad Autonoma Metropolitana, Xochimilco, Mexico City 04960, Mexico; 6Departamento de Biología Molecular e Histocompatibilidad, Hospital General “Dr. Manuel Gea Gonzalez”, Mexico City 14080, Mexico; cvazquez@facmed.unam.mx; 7Laboratorio de Biotecnologia, Departamento de Sistemas Biologicos, Universidad Autonoma Metropolitana, Xochimilco, Mexico City 04960, Mexico; aesquivel@correo.xoc.uam.mx (A.L.E.-C.); jezuniga@correo.xoc.uam.mx (E.Z.-L.); 8Laboratorio de Biologia Experimental, Departamento de Sistemas Biologicos, Universidad Autonoma Metropolitana, Xochimilco, Mexico City 04960, Mexico; fmendoza@correo.xoc.uam.mx; 9Clinica de Obesidad, Hospital General “Dr. Manuel Gea Gonzalez”, Mexico City 14080, Mexico; merr_10mx@yahoo.com.mx; 10Unidad de Soporte Nutricional, Instituto Nacional de Neurologia y Neurocirugia–INNNMVS “Manuel Velasco Suarez”, Mexico City 14269, Mexico; 11Division de Ciencias Biologicas y de la Salud, Departamento de Ciencias de la Salud, Universidad Autonoma Metropolitana Iztapalapa, Mexico City 09310, Mexico; omarperval@gmail.com

**Keywords:** obesity, adipose tissue, low-grade chronic inflammation, cellular immunity, T lymphocytes, B lymphocytes

## Abstract

Obesity is considered a state of chronic low-grade inflammation that impacts the development of chronic degenerative diseases. Cellular immunity plays a crucial role in the onset and persistence of this inflammatory condition. As the degree of obesity increases, significant distinct immunometabolic alterations are observed compared to individuals with normal weight. Moreover, obese patients who undergo bariatric surgical procedures for weight loss exhibit changes in the proportion of immune cells. These alterations help to explain several molecular processes associated with inflammation in obesity, including protein activation and inactivation, precursor molecule synthesis, phosphorylation events, and the activation of signal transduction pathways, all of which are orchestrated by immune cells, primarily lymphocyte subpopulations. The study of the immunometabolic profile through lymphocyte subpopulations in obese patients can provide a more comprehensive and objective understanding of disease severity and the risk of developing obesity-related chronic degenerative conditions and thereby improve or propose therapeutic and novel approaches. Therefore, the objective of this narrative review is to offer an integrative perspective on the molecular and pathophysiological mechanisms through which lymphocyte populations contribute to obesity-related inflammation and how weight loss through bariatric surgical procedures may contribute to the therapeutic management of inflammation.

## 1. Introduction

Obesity is a multifactorial chronic disease characterized by the excessive accumulation of adipose tissue (AT), which manifests as an increase in total body fat (TBF). Commonly attributed to an imbalance between caloric intake and energy expenditure, its pathophysiology results from a complex interplay of genetic, environmental, behavioral, emotional, cultural, socioeconomic, and political factors [1].

AT is not merely a passive reservoir for energy storage; rather, it functions as an active endocrine organ. Its complex and heterogeneous composition includes preadipocytes, mature adipocytes, fibroblasts, endothelial cells, extracellular matrix proteins, mesenchymal stem cells, and various immune cell populations [2,3]. AT is generally classified into white adipose tissue (WAT) and brown adipose tissue (BAT). BAT is primarily involved in thermogenesis and energy expenditure; WAT serves as the main site of energy storage [4]. WAT is further subdivided into subcutaneous (under the skin) and visceral (surrounding internal organs) compartments, with the latter being more closely associated with metabolic risk (Figure 1) [5].

WAT expands through two main processes: adipocyte hypertrophy, characterized by an increase in the cell size, and hyperplasia, which refers to an increase in the number of adipocytes. This compensatory growth initially allows for the management of excess lipids by increasing the tissue’s storage capacity. However, when adipocytes become enlarged and exceed a critical threshold, oxygen delivery to the tissue becomes insufficient, resulting in a hypoxic state, which triggers several pathological changes, including extracellular matrix remodeling, fibrosis, adipocyte necrosis, and lipid release from damaged cells [6].

These alterations activate both innate and adaptive immune cells and stimulate the production of low-molecular-weight signaling molecules, known as cytokine messenger proteins, and glycoproteins that orchestrate immune responses, mediate intercellular communication, and regulate both systemic and local inflammation [7,8]. The term “inflammation” is derived from the Latin word inflammation, meaning “to ignite” or “to set on fire”, and is often identified by the suffix “i”, which denotes a nonspecific bodily response to harmful stimuli. Inflammation initially manifests as an acute physiological process that promotes complete tissue regeneration and the restoration of homeostasis through resolution mechanisms. However, in obesity, this process becomes disrupted, persists, and ultimately develops into chronic inflammation [9,10,11,12,13].

Metabolic dysfunction leads to mutual regulation between immune mechanisms and disrupted metabolic pathways, leading to sustained inflammation even in the absence of pathogens. This state is known as low-grade chronic inflammation, which includes metainflammation-related obesity (MIOR) [9], and it persists when the initiating trigger is not eliminated or when the resolution mechanisms are impaired [14]. WAT serves as the primary site where systemic low-grade chronic inflammation originates. In the context of obesity, lipid accumulation within AT initiates an inflammatory response, resulting in the increased secretion of pro-inflammatory cytokines. These molecules can activate signaling pathways such as C-Jun N-terminal kinase (JNK) and nuclear factor kappa B or kappa-light-chain-enhancer of activated B cells (NF-κβ) in peripheral tissues, including the liver and skeletal muscle, thereby impairing systemic insulin signaling [15]. Particularly, adipose-tissue-resident macrophages are subdivided into two subpopulations [8], vascular-associated and lipid-associated, and the decrease in macrophages in AT is accompanied by a decrease in the pro-inflammatory profile [16,17].

This inflammatory environment contributes to the development and perpetuation of various chronic conditions, including type 2 diabetes (T2D), systemic arterial hypertension, acute myocardial infarction, stroke, certain types of cancer, and metabolic dysfunction-associated steatosis liver disease (MASLD) [14,15]. Conversely, following bariatric surgery (BS), the attenuation of MIOR has been associated with improvements in the immunometabolic profile and reductions in obesity-related morbidity and mortality [16,17].

Therefore, the objective of this review is to offer an integrative perspective on the molecular and pathophysiological mechanisms through which lymphocyte populations contribute to obesity-related inflammation and how weight loss through BS procedures may contribute to the therapeutic management of inflammation.

## 2. Immune Response in Obesity

### 2.1. Cellular Immunity

In obesity cellular immunity plays a central role, as the excessive expansion of AT generates signals that continuously activate leukocytes, thereby disrupting the balance between pro-inflammatory and regulatory immune mechanisms [18,19]. The interaction between immune cells and adipocytes is bidirectional. Hypertrophic adipocytes secrete chemokines such as monocyte chemoattractant protein-1 (MCP-1/CCL2), which recruit monocytes, while immune cells release cytokines that decrease insulin sensitivity in adipocytes [20]. Cellular immune dysfunction in obesity is not restricted to AT; it also alters leukocyte activity in the systemic circulation, liver, skeletal muscle, and pancreatic islets, contributing to insulin resistance (IR), MASLD, and impaired B cell function. Moreover, obesity can compromise immune memory. T and B lymphocytes exposed to a metabolically stressful environment exhibit alterations in memory formation, leading to deficient adaptive responses [21].

### 2.2. The Basis of the Inflammatory Process

Inflammation is a concept rooted in ancient medicine, historically defined by hallmark features such as swelling, redness, heat, pain, and impaired function (e.g., stiffness or restricted mobility). Today, it is understood as a dynamic and multifaceted biological response to tissue damage, triggered by harmful stimuli like chemical toxins, environmental stressors, physical trauma, repetitive strain, or pathogens [22], involving a complex sequence of events aimed at eliminating insults and restoring homeostasis [23]. This complex sequence of events includes the following: the initiation of and recognition between tissue-resident sentinel cells—macrophages, dendritic cells, and mast cells—detect pathogen-associated molecular patterns (PAMPs) or danger-associated molecular patterns (DAMPs). This is achieved via pattern recognition receptors, such as toll-like receptors (TLRs) and nucleotide-binding oligomerization domain-like receptors or NOD-like receptors (NLRs). This step triggers the release of pro-inflammatory cytokines (e.g., tumor necrosis factor alpha [TNF-α] and interleukin-1 beta [IL-1β]), vasoactive amines (e.g., histamine), and lipid mediators (e.g., prostaglandins) that increase the vascular permeability and leukocyte recruitment (Figure 2A). Afterwards, the phase in which endothelial activation leads to leukocyte rolling, adhesion, and extravasation starts (Figure 2B). Later the removal of the injurious agent by recruited leukocytes, especially neutrophils and macrophages, eliminates pathogens and cellular debris though mechanisms such as phagocytosis, the production of reactive oxygen species (ROS), and the release of antimicrobial peptides [11] (Figure 2C). Then, the regulation of the inflammatory response begins to prevent excessive tissue damage. This is achieved through the secretion of anti-inflammatory cytokines (e.g., IL10, transforming growth factor beta [TGF-β]) and specialized pro-resolving mediators (SPMs) (e.g., resolving, protecting). SPMs are produced endogenously from omega-3 polyunsaturated fatty acids (PUFAs) derived from membrane phospholipids. SPMs act by binding to G protein-coupled receptors, controlling tissue homeostasis and inflammation by restricting the neutrophil invasion into inflammatory foci, enhancing the cytotoxicity of apoptotic cells, decreasing the production of inflammatory cytokines, and promoting anti-inflammatory macrophages’ polarization (M2). M2 partly prevents the transition of inflammation to chronicity by stimulating neutrophil apoptosis [12,13] (Figure 2D). Once the harmful stimulus is eliminated, the tissue enters the resolution phase. This includes the regeneration of parenchymal cells (if possible), the deposition of extracellular matrix components, angiogenesis, and the remodeling of the tissue architecture to restore function [24] (Figure 2E).

These inflammatory responses can support tissue repair and immune defense, but under certain conditions, they may also contribute to the progression of chronic diseases. Functionally, inflammation serves as a crucial component of the body’s secondary line of defense against microbial invasion [22].

### 2.3. Mechanism of Low-Grade Chronic Inflammation Associated with Obesity

In obesity, inflammation is systemic, low-grade, and primarily T lymphocyte-dependent. It occurs when AT and other insulin target tissues inhibit elevated levels of inflammatory factors and infiltrating immune cells, such as T helper lymphocytes (CD4+) which secrete cytokines [15]. This form of low-grade chronic inflammation does not necessarily involve tissue damage or structural alteration, as is commonly observed in classic inflammatory responses. Immune cell activation in obesity is primarily triggered by DAMPs, such as free fatty acids, low-density lipoprotein cholesterol (LDL-c), uric acid, high-mobility group box 1 (HMGB1) protein, mitochondrial deoxyribonucleic acid (DNA), and extracellular adenosine triphosphate (ATP). This mechanism contrasts with infection-induced inflammatory responses, which are predominantly activated by PAMPs [15,27,28,29]. Clinical obesity has been defined as a chronic disease resulting from dysfunction in one or more organs or systemic alterations, directly induced by excess AT, which may lead to life-altering or life-threatening complications [30]. Under normal physiological conditions, adipocyte hypertrophy and hyperplasia stimulate angiogenesis to ensure an adequate supply of nutrients and oxygen to the expanding AT. However, when the AT expansion occurs rapidly or exceeds its remodeling capacity, the oxygen demand cannot be fully met, promoting local hypoxia in adipocytes [31] and leading to metabolic stress. This condition activates and induces the accumulation of hypoxia-inducible factor 1 (HIF-1) [32] (Figure 3).

Adipocyte normoxia: HIF-1α hydroxylated by two proline residues is associated with the Von Hippel–Lindau (VHL) protein and is tagged with ubiquitin, leading to its degradation via the proteasome (Figure 3A). In hypoxic conditions in the adipocyte and immune cells within AT, HIF-1α translocates to the nucleus and associates with hypoxia-inducible factor 1 beta (HIF-1β) [32]. This complex then binds to the hypoxia response element (HRE) region of DNA, resulting in the transcription of genes involved in processes such as angiogenesis, erythropoiesis, cell proliferation and survival, autophagy, apoptosis, glycolysis, and macrophage polarization (Figure 3B) [32]. HIF-1 may be involved in the expression of genes encoding NF-κβ (induced by hypoxia). Then subunits’ NF-κβ can then translocate to the nucleus to activate genes of cytokines, chemokines, or membrane receptors, thus leading to the proliferation of B lymphocytes, natural killer (NK), dendritic, and T helper cells and cytotoxic lymphocytes, which help modify the number and activation states of macrophages (Figure 3C) [32]. Also, NF-κβ regulates the quantity of HIF-1 under hypoxic conditions (Figure 3D).

In macrophages, the presence of a stimulus such as TNF-α (secreted by hypoxic adipocytes) activates its Ikappaβ kinase (IKK) complex, which is mediated by the phosphorylation of IkappaB (Iκβ) protein, making it suitable for proteasomal degradation (Figure 2E). This action results in the release of NF-κβ subunits and their translocation to the nucleus to activate target genes [33,35].

Two apoptotic pathways have been identified in adipocyte death: (1) the intrinsic pathway, mediated by the release of cytochrome C from the mitochondria into cytosol, and (2) the extrinsic pathway, mediated by death receptors (primarily TNF-α) and exclusively regulated by caspases (CASPs) [35]. The activation of specific CASPs may also lead to pyroptosis (a form of programmed cell death distinct from apoptosis) characterized by caspase 1 (CASP1) and caspase 3 and 7 (CASP3/7) activation, by pro-inflammatory cytokine release, and by rupture of the plasma membrane [35]. Adipocyte death has been identified as a key initiating event in low-grade chronic inflammation [36,37,38].

The macrophages polarize into M1 (classically activated) or M2 (alternatively activated) phenotypes. M1 secretes pro-inflammatory cytokines such as interferon-gamma (IFN-γ), interleukins (IL-1, IL-12, and IL-18), and TNF-α, initiating the inflammatory response [39]. M2 conversely releases mediators such as TGF-β, IL-4, IL-10, and IL-13 and lipid resolution mediators [40,41].

Upon entry into the hypoxic environment of the expanding AT, macrophages accumulate HIF-1α, promoting their polarization to the M1 phenotype. This results in prolonged macrophage survival (through apoptosis inhibition) and the increased secretion of ROS and pro-inflammatory cytokines into AT [42]. By expanding AT, M1 encircles necrotic or apoptotic adipocytes (which release DAMPs) and forms crown-like structures to eliminate them. M1 also generates inflammasome (NLRP3) and apoptotic activity in response to DAMPs [43,44]. This means that the innate immune system remains in a state of pre-activation, which predisposes it to infections [42].

Also, live adipocytes in hyperplasia and hypertrophy alone are capable of secreting pro-inflammatory and prothrombotic factors such as IL-1β, IL-6, TNF-α, leptin (adipokine), chemokines, macrophage colony-stimulating factor, MCP-1/CCL2, C-reactive protein (CRP), tissue factor VII, and plasminogen activator inhibitor type 1 (PAI-1) [45,46]. These factors also support M1 differentiation and survival, and they also promote greater monocyte and myeloid cell infiltration into AT [4,47,48].

Additional mechanisms implicated in MIOR include the following:–Saturated fatty acids released by apoptotic adipocytes (recognized as DAMPS) promote M1 activation via indirect binding to toll-like receptor 4 (TLR4) and toll-like receptor 2 (TLR2). This leads to NF-κβ and JNK activation, resulting in the secretion of pro-inflammatory cytokines such as IL-1β, MCP-1, and TNF-α [34,48].–Hypoxia and hypertrophy adipocytes may contribute to mitochondrial dysfunction, leading to the accumulation of intracellular fatty acids and their metabolites (e.g., fatty acyl-Coenzyme A and diacylglycerols), resulting in lipid peroxidation and oxidative stress. This is characterized by increased ROS production (particularly superoxide ions and nitric oxide) and the further recruitment of immune cells to AT. Elevated levels of TNF-α and leptin, along with reduced levels of IL-10 and adiponectin (anti-inflammatory molecules), have also been reported [45,47] (Figure 4).

### 2.4. Cancer and Low-Grade Chronic Inflammation Associated with Obesity

#### 2.4.1. Low-Grade Chronic Inflammation Associated with Obesity

Although alterations in the immune system are frequently observed in individuals with obesity, it remains unclear whether impaired immune surveillance plays a direct role in increasing their vulnerability to cancer [48]. Inflammation in AT, particularly WAT, leads to chronic tissue injury that triggers wound healing mechanisms and establishes a pro-neoplastic microenvironment. Injured WAT represents a rich source of pro-inflammatory mediators, including increased leptin, visfatin, CRP, IL-6, MCP-1/CCL2, insulin-like growth factor-binding protein 2 (IGFBP-2), TNF-α, IL-1β, cyclooxygenase-2 (COX-2)-derived prostaglandin E2, serpin 1, matrix metalloproteinase 2, CCL-20, leukemia inhibitory factor, and chemokines from the chemokine (C-X-C motif) ligand (CXCL) subfamily. The formation of crown-like structures mediated by adipocytes, macrophages, and T and B lymphocytes promotes cancer cell survival and proliferation [49,50,51].

Low-grade chronic inflammation related to obesity, along with mitochondrial dysfunction and endoplasmic reticulum stress, promotes oxidative stress, which induces DNA damage and genomic instability, thereby facilitating cancer development [50]. Moreover, obesity alters NK cell antitumor responses, as the peroxisome proliferation-activated receptor [PPAR] (signaling—critical in lipid metabolism) drives metabolic paralysis in NK cells [52]. Additionally, leptin-induced PD-1 upregulation promotes T cell exhaustion [53], mainly in cytotoxic T lymphocytes (CD8+ or CTLs). This exhaustion is characterized by an increased expression of immune checkpoints (PD-1, LAG-3, Tim-3) and the inability to proliferate and produce effector molecules such as IFN-γ [54].

#### 2.4.2. Consequences and Complications of Chronic Inflammation Associated with Obesity

Chronic inflammation associated with obesity influences the efficacy of immunotherapy. Notably, the immune checkpoint blockade (ICB) has improved outcomes in several cancers; however, a major limitation is that most patients fail to respond, partly due to immunosuppression. Interestingly obesity appears to enhance the efficacy of these therapies in certain types of cancer, such as breast cancer (BC). In lean and obese murine models treated with anti-PD-1, tumor regression occurred in lean mice, whereas obese mice exhibited a potent inhibition of tumor progression, suggesting that obesity modulates therapeutic responses [55].

Remarkably, anti-PD-1 increased antitumor immunity even within the immunosuppressive obese state, restoring a lean-like immune phenotype [56]. Nevertheless, obesity also suppresses tumor-infiltrating CD8+ T lymphocytes, thereby accelerating malignant proliferation, metastasis, and the deterioration of the antitumor immune response [57]. Furthermore, this inflammation contributes to hyperinsulinemia, which enhances glycolytic activity, generating oxidative stress and consequent DNA damage. Hyperinsulinemia also potentiates signaling through the PI3K pathway, thereby promoting tumorigenesis [50]. IR stimulates the synthesis of IGF-1, a potent mitogenic factor that activates the PI3K/Akt/mTOR and Ras/MAPK pathways, driving tumor growth [51].

Another critical aspect is obesity-induced gut microbiota dysbiosis, which disrupts metabolic and immune signaling, promoting metabolic endotoxemia. This condition triggers systemic IL-6 production via TLR5, thereby fostering tumor growth [58]. Consequently, obesity represents an increased risk and worse outcomes across various cancer types. However recent epidemiological studies have reported that overweight and obesity are associated with improved survival during and after cancer treatment, a phenomenon known as the “obesity paradox” [59]. This paradox suggests that excess energy reserves may provide survival advantages during acute hypercatabolic cancer phases, although it remains controversial due to methodological heterogeneity, confounding variables, and interpretation biases [60].

## 3. Cellular Immunity and Its Participation in Obesity

### 3.1. Adipose Tissue’s Immune Cells

This section will emphasize the lymphoid cells that participate in MIOR (mainly T lymphocytes).

Macrophages are not the only immune cells responsible for initiating and sustaining inflammation within AT. A variety of immune cell populations actively contribute to this process, enabling inflammation that initially arises in AT to propagate systemically [3].

T lymphocyte CD3+—characterized by a Cluster of Differentiation (CD) 3 surface expression, a transmembrane marker associated with the T cell receptor (TCR)—produces cytokines that directly influence macrophages and adipocytes in the AT of obese individuals. These cells represent one of the most abundant immune populations in AT and are subclassified by their surface markers into CD4+ or CD8+. Upon interacting with the major histocompatibility complex (MHC), T cells become activated effector cells. These are thought to accumulate in VAT via CCR5-CCL5 interactions, a chemokine superfamily involved in inflammatory responses and in promoting the adhesion and migration of various T cell subsets during immune activation [1]. An increase in the production of pro-inflammatory cytokines by these T cells activates signaling pathways mediated by JNK and the inhibition of Iκβ kinase (IkβK) in adipocytes and insulin target cells such as myocytes and hepatocytes [61]. These pathways induce the transcription of genes encoding pro-inflammatory proteins, which in turn inhibit insulin receptor substrate 1 (IRS-1) through serine–threonine phosphorylation, resulting in IR. This response is thought to restrict the glucose uptake by other tissues, thereby prioritizing glucose availability for the immune system to use during inflammation [47]. Elevated levels of TNF-α in the AT of obese individuals play a direct role in IR. TNF-α activates the NF-κβ intracellular signaling cascade, which inhibits the phosphorylation of insulin receptor substrate 2 (IRS-2), serine/threonine kinase or protein kinase B (AKT/PKβ), glucose transporter 2, and glucose transporter 4, culminating in IR and hyperglycemia [62]. The production of these cytokines also activates NF-κβ, which increases the genetic expression to produce more cytokines like IL-6, TNF α, IFN-γ, TGF-β, MCP-1, and IL-1β, which are transported through the bloodstream, participating in the inflammation of body parts other than AT [47].

CD4+ T lymphocytes recognize presented antigens and macrophages via the MHC class II. These cells differentiate into two subsets: T help 1 (Th1) and T help 2 (Th2). Th1 cells are primarily involved in cellular immune responses, playing a critical role in chronic inflammation, thereby contributing to the development of IR and metabolic syndrome (MS) [39]. Effector T helper lymphocytes (CD4+CD62−), a subclass of CD4+ T cells with effector functions and a Th1 profile, modulate both the number and activation status of macrophages within AT [62,63], primarily through the secretion of IFN-γ, TNF-α, and IL-6 [64].

CD8+ T lymphocytes recognize antigens in MHC class I cells. These cells differentiate into cytotoxic T cell 1 (Tc1) and cytotoxic T cell 2 (Tc2), with TC1 exhibiting a secretion pattern like Th1, including IFN-γ. Within VAT, CD8+ can activate into effector cytotoxic T lymphocytes (CD8+CD28−), an early event in obesity associated with inflammatory responses that promotes macrophage recruitment and differentiation. As such, they play an essential role in the initiation and maintenance of AT inflammation and in the systemic progression of comorbidities such as IR [64]. The stimulation of T lymphocytes by IL-15, a pro-inflammatory cytokine elevated in individuals with high-fat diets, obesity, and fatty liver disease [65], enhances the expression of the Carnitine palmitoyl transferase 1A gene (CPT1A), which promotes fatty acid oxidation, a crucial process providing energy for the proliferation and survival of CD8+ T cells [66].

On the other hand, both CD4+ and CD8+ T cells can be further classified into naive T cells (Th0) and memory T cells. Notably, Th0 cells have been reported to promote chronic systemic inflammation, increasing significantly with aging and obesity [67,68,69]. Furthermore, chronic inflammation in obese patients accelerates the telomere shortening in memory T cells, predisposing these individuals to an elevated risk of age-related diseases and chronic inflammatory conditions. Premature aging fosters the generation of dysfunctional mitochondria, leading to the production of ROS and the activation of NF-κβ, which in turn perpetuates MIOR. Consequently, individuals with obesity often exhibit accelerated aging compared to those with a normal body weight [70]. Severe obesity accompanied by sarcopenic obesity is associated with endocrine disorders, premature aging, and reduced physical activity, which may also influence Th0 and memory T cell dynamics [71].

B lymphocytes are among the earliest immune cells to infiltrate AT, primarily VAT [72,73]. Once within AT, B cells produce pro-inflammatory mediators that regulate inflammatory T lymphocytes and macrophages while secreting autoantigen-specific immunoglobulin G (IgG) antibodies targeting adipocytes [74,75]. Additionally, they modify the number and activation status of M1 through the release of IFN-γ, TNF-α, and IL-6 [62,63,64]. Notably, B cell function declines with obesity and aging, a phenomenon linked to diminished responses to infections and vaccinations [62,75,76,77,78].

The functionality of NK cells, characterized by the surface expression of CD16+ and CD56+, is impaired in obesity, compromising their cytotoxic capacity, and interfering with immune surveillance against tumor cells and virus-infected cells. This dysfunction is associated with a chronic inflammatory environment, marked by elevated levels of cytokines such as IL-6, TNF-α, and IL-1β, as well as the disrupted signaling of activating receptors such as natural killer group 2 member D (NKG2D). This transmembrane activating receptor, expressed by NK cells, recognizes self-proteins that appear on the surface of stressed, malignant, or infected cells [1]. Additionally, IFN-γ production further exacerbates NK cell dysfunction. Moreover, inverted free fatty acids and advanced glycation end products contribute to metabolic dysfunction, affecting mitochondrial biogenesis and iron metabolism in these cells, which further worsens their impairment in obesogenic environments [79].

### 3.2. Endothelial Dysfunction in Obesity

“Endothelial dysfunction” is characterized by damage to the vascular endothelium, which leads to alterations in its normal functions, such as regulating the blood flow, vascular tone, immune response, and coagulation processes. This dysfunction is mainly manifested by the onset of vascular stiffness, the decreased production of nitric oxide (an essential vasodilator molecule), and the formation of blood clots, which increases the risk of atherosclerosis and other cardiovascular diseases. Thus, endothelial dysfunction becomes a crucial link between obesity, inflammation, and cardiovascular risk [80]. In individuals with severe obesity, endothelial dysfunction is exacerbated by high concentrations of LDL-c and reduced levels of HDL-c [67], the dysregulation of endocrine and paracrine effects derived from AT inflammation and IR [80].

An increase in circulating CD8+ cells has been associated with the development of endothelial dysfunction by regulating monopoiesis and macrophage accumulation during early atherosclerosis, exerting cytotoxic activity within atherosclerotic plaques, promoting macrophage apoptosis, and facilitating necrotic core formation [67].

MIOR disrupts vascular homeostasis and serves as a key contributor to endothelial dysfunction by promoting the production of acute-phase proteins, pro-inflammatory cytokines (e.g., IL-6 and IL-1), and TNF-α–derived microparticles [81], as well as nicotinamide adenine dinucleotide phosphate (NADP) oxidase in perivascular adipose tissue (PVAT) and M2 (upon interaction with oxidized LDL-c). TNF-α activates NF-κβ/JNK signaling, whereas IL-6 activates Janus kinase and the signal transducer and activator of transcription (JAK/STAT3); both cytokines increase ROS production, which reduces inducible nitric oxide synthase expression, promoting vasoconstriction, leukocyte adhesion, and vascular permeability. In cardiometabolic models, the paracrine TNF signaling from smooth muscle to endothelial cells elevates blood pressure in obesity. Therefore, monitoring these biomarkers associated with endothelial dysfunction and understanding the role of CD8+ T lymphocytes in atherosclerosis may pave the way for the early detection of pathological conditions and even inform novel therapeutic approaches [82].

### 3.3. Cellular Immunity as a Key Factor in Inflammation Amplification

The enhancement of MIOR appears to be the primary factor influencing leukocyte numbers in circulation. Lymphocyte subpopulations increase in peripheral blood, amplifying inflammatory responses and playing a crucial role in the initiation of obesity-associated inflammation [83,84,85].

The infiltration of M1 into AT and myocytes also contributes to MIOR [85,86]. Likewise, obese individuals exhibit the increased infiltration and activation of Kupffer cells (specialized resident hepatic macrophages) [87], which promote lipogenesis and the biosynthesis of toxic ceramides. This process enhances the production of insulin-like growth factor binding protein-7 (IGFBP7), which directly impairs insulin receptor signaling in the liver, leading to IR and metabolic-associated MASLD [43,88]. The accumulation of energy in the form of triglycerides within adipocytes promotes their hypertrophy and hyperplasia, leading to structural changes in AT, particularly in VAT. These alterations result in the infiltration and activation of leukocyte populations. Immune cells, along with adipocytes, secrete cytokines that primarily induce local damage, notably IR. Once these cytokines reach systemic circulation, they contribute to the inflammation of other organs, such as skeletal muscle and the liver, exacerbating systemic IR and disrupting energy metabolism. This metabolic impairment initially manifests as MS and subsequently progresses to additional comorbidities associated with obesity [31,34] (Figure 5).

### 3.4. Evidence of Changes in Lymphocytes in People with Obesity

Studies in individuals with obesity and/or MS have shown that the concentration of leukocyte populations in the bloodstream differs significantly from that of their lean counterparts. Increased absolute leukocyte counts (cells/µL) have been reported [88,89].

Elevations in NK cells (%) have been positively associated with: MS, visceral fat (VF), skeletal muscle mass, homeostatic model assessment of insulin resistance (HOMA-IR) and dietary carbohydrate intake in patients 18–60 years old, whose body mass index (BMI) was greater than 40 kg/m^2^, and who were candidates for BS [88,89,90,91].

Total lymphocyte counts (% and cells/µL) are positively associated with: VAT’s accelerated T lymphocyte differentiation (% and cells/µL), and CD8+ T cell differentiation. Patients with MS had significantly lower CD4+ RTL than patients without MS. Within the first 6 months after BS, the relative telomere length increased in CD4+ T cells, after which it decreased [68,85,91,92,93]. Increased levels of CD4+ T cells are positively associated with CRP and IR in women and men of 35–65 years with a BMI of 25–35 kg/m^2^. Which strongly supports the role of the adaptive immune system in mediating obesity-induced inflammation and IR. Additionally, elevated counts of CD4+ and CD4+CD62− (cells/µL); CD8+ (cells/µL); CD8+CD28− (% and cells/µL); and naive T helper lymphocytes (CD4+CD45RA+) (% and cells/µL) have been positively correlated with fat-free mass (FFM) [66,68,69,94,95].

An increase in memory T helper lymphocytes (CD4+CD45RO+) (% and cells/µL) [66,69,91] has shown a negative association with the FFM percentage in Mexican women and men > 18 years old with different degrees of obesity. Moreover, higher levels of naive cytotoxic T lymphocytes (CD8+CD45RA+) (cells/µL) and memory cytotoxic T lymphocytes (CD8+CD45RO+) (% and cells/µL) [68] are positively associated with the BMI and waist circumference. Furthermore, an increase in B lymphocytes (% and cells/µL) [94,95], senescent B cells, and double-negative B cells has been observed in both circulating blood and AT, showing an increase in the levels of ROS, the phosphorylated Adenosine monophosphate-activated protein kinase (AMPK), and Sestrin 1 are both able to mitigate stress and cell death [96,97].

Conversely, other studies have reported lower counts of the following immune cells in patients with obesity and/or MS compared to their lean counterparts: a decreased percentage of total lymphocytes [69], CD4+CD45RA+ [91], CD8+CD45RA+, and B cells (% and cells/µL) [68,75]. Additionally, a lower percentage of memory B lymphocytes in circulation compared to AT has been observed. In the bloodstream, these cells are metabolically less active in terms of pro-inflammatory substance production and the expression of enzymes involved in glucose oxidation metabolism, compared to their counterparts in AT [97].

Given this panorama of alterations in cellular immunity during obesity, it is important to compare the changes that have occurred to people who no longer have obesity due to weight loss [98,99].

## 4. Changes in Cellular Immunity in Obese People Undergoing Bariatric Surgery

Obesity treatment involves lifestyle modifications through a diet plan, exercise, and behavioral management. In some cases, weight loss medication can be effective; however, in the presence of severe obesity that has chronically perpetuated the inflammatory state, BS offers an additional treatment option when previous approaches have failed. A laparoscopic Roux-en-Y gastric bypass (LRYGB) and laparoscopic sleeve gastrectomy (LSG) are valuable models for understanding and comparing changes in cellular immunity [100]. Surgical procedures such as BS represent an effective treatment for obesity and T2D, restoring metabolic homeostasis through the normalization of bile acids and the microbiome gut–brain axis [101]. The comparison of the immune profile in obese individuals before and after undergoing metabolic BS reveals significant changes. In general, BS has been reported to reduce pro-inflammatory markers, including HOMA-IR, LDL-c, CRP, IL-1β, IL-12, IL-18, and IFN-γ [102], as well as the expression of PAI-1, intercellular adhesion molecule-1 (ICAM-1), TNF-α, resistin, leptin, IL-6, IL-1, S100 calcium-binding protein A9, NF-κβ, TLR4, TLR2, CD14 membrane receptor, matrix metalloproteinase-9, and MCP-1 [94,102,103,104,105,106,107,108,109,110,111]. Additionally, BS has been linked to a reduction in thrombomodulin, prothrombin time, neopterin, and lactoferrin, with the latter two recognized as biomarkers of endothelial inflammation [110,111,112]. There are also significant changes in cellular immunity among different BS techniques.

### 4.1. Restrictive Bariatric Surgery Techniques

Restrictive BS techniques reduce the stomach volume, inducing weight loss by limiting food intake [113]. In individuals undergoing LSG a significant increase in CD8+, CD8+CD28− [114] memory T lymphocytes [115], and regulatory T cells (Tregs) has been observed, with blood concentrations like those in lean controls. However, the Tregs restoration is less complete than in lean individuals, while IL-10 levels (an anti-inflammatory cytokine) are elevated [116].

Conversely, LSG has been associated with a significant reduction in blood counts of total leukocytes; NK cells [116]; and T lymphocyte subtypes Th1, Th17, and Th2 (%), reaching levels comparable to lean controls [113]. Additionally, a decrease in CD4+ (% and cell/µL) [116] has been linked to a BMI reduction [117]. Reduced levels of CD4+CD45RO+ and B lymphocytes, correlated with lower IgG and TBF levels [103], have also been observed. Furthermore, the decrease in CD8+CD28− [116] following weight and BMI reductions has been associated with a decreased hepatic expression of Tsukushi (TSK), a heptatonic, thereby lowering the risk of metabolic diseases [105,116].

In laparoscopic greater curvature plication and laparoscopic gastric banding (LGB), a decrease in CD4+ T cells has been reported, linked to weight and BMI reductions [117,118]. Additionally, LGB has been associated with a significant increase in total circulating CD3+ T cells, correlated with BMI reduction [117].

### 4.2. Mixed Bariatric Surgery Techniques

Mixed BS techniques not only reduce the stomach volume, thereby limiting the food intake, but also cause nutrient malabsorption, as nutrients are eliminated through feces since they bypass absorptive and secretory regions of the stomach and the small intestine.

LRYGB has been reported to significantly increase cytotoxic NK cell activity in the blood [119], the natural killer T cell percentage, and CD3+ counts [108], which are associated with a BMI reduction [117]. Other increases include CD8+ T cells [101]. Weight loss in individuals with obesity has been associated with an improved antitumor response of CD8+, indicating that the impairment of the antitumor immune response caused by obesity can be partially reversed through weight loss [57]. Th0-inducing IL-17 [106,120], central memory T cells, and Tregs also increased [105]. Additionally, improved follicular T helper cell function has been reported, leading to a decrease in INF-γ, IL-2, IL-4, and IL-17 secretion. This, in turn, promotes the development of regulatory B cells (Breg) and the modulation of their programmed cell death [106]. Further reports indicate an increase in the B cell percentage [116], memory B cells [106], immunoglobulins IgG; IgA; and IgM, as well as adiponectin, IL-12, and IL-8, reflecting enhanced immune system functionality [103,108,109,117].

On the other hand, LRYGB has been associated with a significant reduction in the total leukocyte count and total CD3+ [89,104,116,120,121,122]. These reductions have been linked to greater weight loss and BMI reductions [118,121]. Additionally, decreased CD4+ (% and cells/µL) and CD4+CD62− (cells/µL) [101] correlated with weight loss, BMI reduction [117,118], VF, glycated hemoglobin (HbA1c) [115], and IR [90,92], suggesting a reduction in inflammatory activity mediated by these cells and, consequently, low-grade chronic inflammation [116,123]. Moreover, decreased CD4+CD45RA+ (% and cells/µL) [92,116] has been linked to reduced MS [92]. This may be attributed to the lower dietary calorie intake and weight loss due to LRYGB, which reduces the overstimulation of the interleukin-7 receptor (IL-7R) and, consequently, IL-7 levels, a cytokine that, in animal models, has been shown to regulate naive T cell homeostasis [124,125].

Additionally, reductions in CD4+CD45RO+, CD8+, CD8+CD28−, and CD8+CD45RO+ (% and cells/µL) have been correlated with decreased MS [92,116], further suggesting improvements in obesity-related inflammation [68].

In patients undergoing laparoscopy Roux-en-Y an increase or no significant change in the percentage of B cells have been observed [106,116,123] along with a decrease in their activity [123,126,127]. For instance, the reduced production of autoreactive IgM and IgG antibodies has been documented, which was associated with decreased TBF, glucose levels, and HOMA-IR. The decline in IgG activity has been partly attributed to weight loss, as both caloric restriction and nutrient deficiencies induced by BS may contribute to a certain degree of immunodeficiency [102]. However, changes in B cell function may also be explained by the modulation of pro-inflammatory T lymphocytes, which promote a shift in B cells from an effector phenotype [128] to a regulatory phenotype [106]. This shift leads to increased production of IL-10 and TGF-β, associated with reductions in the BMI and TBF, as well as the decreased secretion of INF-γ, IL-2, IL-4, IL-6, and IL-17 [129]. In individuals undergoing laparoscopic biliopancreatic diversion (LBD), the improved B cell-mediated production of IgG, IgA, and IgM antibodies has been reported, along with decreased circulating levels of NK cells, CD8+ T cells, and B cells (both % and cells/µL) [89].

### 4.3. Differences in Peripheral Immune Cell Profiles Between the Two Most Used Bariatric Surgical Techniques (Gastric Bypass and Sleeve Gastrectomy)

In LSG, a significant reduction in CD4+CD62− T cells has been reported, compared to LRYGB [102]. Other studies, however, have found no significant differences between these procedures [17,107]. Overall, LRYGB appears to induce more pronounced changes in immune cell populations and is currently considered the most effective BS technique for reducing chronic inflammation [104,116,118].

## 5. New Perspectives

Recent advances have significantly expanded our understanding of MIOR, revealing that it is not merely a consequence of adipocyte hypertrophy or cytokine dysregulation, but rather a multifactorial immunometabolic disorder involving the following (Figure 6): he modulation of the inflammatory response through diet and nutraceutical interventions (Section 5.1); the metabolic reprogramming of immune and senescent cells via pharmacological interventions, weight loss, and BS (Section 5.2); the immunomodulation of the gut microbiota (GM) (Section 5.3); and regenerative approaches with AT and adipose-derived stem cells (ADSCs) for the transplantation for metabolic and immune modulation (Section 5.4). These findings point toward novel biomarkers and therapeutic targets, as well as redefine treatment strategies. Below, we address some of these approaches. 

### 5.1. The Modulation of the Inflammatory Response Through Diet and Nutraceutical Interventions

Caloric restriction combined with diets rich in anti-inflammatory bioactive compounds such as ω-3 fatty acids, polyphenols, magnesium, zinc, selenium, fat-soluble vitamins (A, C, D, and E), dietary fiber, and probiotics such as *Lactobacillus paracasei* HII01 helps reduce low-grade chronic inflammation [127].

Curcumin can help reduce TNF-alpha and CRP and improve the lipid profile, IR, and glucose levels [130,131]. This occurs by increasing circulating levels of irisin and adiponectin, activating peroxisome proliferator-activated receptor γ (PPARγ), suppressing neurogenic locus notch homolog protein 1 (NOTCH1) signaling, and regulating the target genes of sterol regulatory element-binding proteins (SREBPs) [132]. Supplementation with curcumin can also help reduce MIOR [133] and promote immunomodulation by improving the GM [134].

Resveratrol has been shown to have anti-inflammatory and immunomodulatory effects through the inhibition of NF-κβ; the activation of extracellular signal-regulated kinases (ERKs) and p38 mitogen-activated protein kinases (MAPKs); TLR-dependent signaling; NLRP3; ROS production; leukocyte infiltration in AT; and the release of pro-inflammatory cytokines [135]. It can also help improve IR through the activation of nicotinamide adenine dinucleotide (NAD^+^)-dependent sirtuin-1 deacetylase (SIRT1) and the inhibition of NF-κβ [136,137].

Berberine can help reduce fasting and postprandial glucose levels, HbA1c, IR, LDL-c, BMI, and the abdominal circumference through its glycolytic activity, as it facilitates insulin release. At the same time, it suppresses hepatic gluconeogenesis and adipogenesis [138]. As an anti-inflammatory agent, it has been observed to decrease the expression of cyclooxygenase-2 and prostaglandin E2, adipocyte differentiation, and the secretion of leptin and resistin, while increasing adiponectin mRNA expression [139].

However, the clinical application and dosing of curcumin, resveratrol, and berberine in humans needs to be standardized, with rigorous evaluations of drug–nutrient interactions and long-term safety studies, so that their incorporation into an integrative medicine framework can provide a more holistic and evidence-based approach to obesity and associated diseases [136,137].

### 5.2. Metabolic Reprogramming of Immune and Senescent Cells

Therapeutic interventions with senolytics (which promote the clearance of senescent cells), senomorphics (which suppress the senescence-associated secretory phenotype [SASP] of senescent T cells without inducing cell death), and autophagy inducers (which stimulate cells to recycle their components and eliminate senescent T cells) could potentially reverse the dysfunctional state in individuals with obesity known as “adipaging”. This condition is characterized by the accumulation of senescent T cells and their SASP, the latter acting as an amplifier of systemic inflammation and IR [140,141,142].

Treatment with metformin aims to modulate fine-tuned immunometabolic pathways, such as inhibiting the overactivation of aerobic glycolysis and the mammalian target of rapamycin (mTOR), promoting the activation of AMPK, enhancing fatty acid oxidation, and stimulating oxidative phosphorylation. These mechanisms can potentially influence the functional differentiation of T cell subpopulations, thereby attenuating MIOR [140,143,144,145,146].

The use of novel NLRP3 inflammasome inhibitors (such as 1,3,4-oxadiazol-2-one derivatives and a sulfonylurea-based inhibitor of NLRP3 [INF200]) has been shown in animal models to reduce IL-1β release, pyroptosis, systemic inflammation, and cardiometabolic complications [147].

The pharmacological induction of autophagy pathways appears to shift macrophage phenotypes toward immunosuppressive resolution states, restoring metabolic flexibility and dampening obesity-induced low-grade chronic inflammation [148].

The “metabolic reprogramming” of T lymphocytes through weight loss can achieve the following:Enhance oxidative efficiency and reduce the dependence on lactate-induced glycolysis, thereby decreasing the activation of pro-inflammatory Th1 and Th17 phenotypes, while increasing regulatory Tregs in AT [140,147].Reverse the dysfunctional phenotype of T cells, characterized by metabolic exhaustion, reduced cytotoxicity, and impaired tumor recognition [48].Achieve the downregulation of Janus kinase 3 (Jak3), a tyrosine kinase associated with cytokine receptors in the intestinal epithelium [148], NF-κβ signaling mediated by TLRs, and the phosphoinositide-3-kinase-protein kinase B (PI3K-Akt) axis (a pathway controlling insulin receptor signaling) [149], suggesting potential therapeutic strategies targeting metabolic signaling in these cells.

In contrast, recent evidence suggests avoiding body weight fluctuations, which can reduce “trained immunity” [innate immune cells have sustained changes in their genetic expression and physiology of innate immune cells, but that does not imply permanent genetic changes (such as mutations and recombination, essential for adaptive immunity)]. In this context, constant body weight fluctuations can cause the TLR4-dependent stimulation of AT macrophages. In innate immune cells fluctuations promote glycolysis and oxidative phosphorylation, increasing TNF-α and IL-6 secretion [149,150].

BS can reverse the biological age of the immune system, thereby reducing the pool of “exhausted” and senescent T lymphocytes. This mitigates “inflammaging”, a concept describing the dual impact of low-grade chronic inflammation [142,151]. 

### 5.3. The Modulation of the Gut Microbiota

Gut dysbiosis, here referred to as the obesogenic gut (OGM), helps perpetuate MIOR through several mechanisms:The alteration of the production, metabolism, signaling, and epigenetic modulation of short chain fatty acids (SCFAs). On one hand, the impaired epigenetic modulation of SCFAs changes the expression of inflammatory genes (via histone deacetylase inhibition and acetylation of NF-κβ target genes), thereby suppressing immune cell activation in AT [152]. On the other hand, the deficient binding/activation of SCFAs to G-protein-coupled receptors (GPCRs) and free fatty acid receptors (FFARs) leads to the inappropriate regulation of GPCR43 (expressed in WAT) and FFAR2/FFAR3 coupled to GPCRs. This results in an increased appetite (due to insufficient intracellular Ca^2+^ elevation to stimulate glucagon like peptide-1 [GLP-1] secretion), decreased energy expenditure, and elevated glucose and insulin levels [143,149].A reduction in butyrate levels decreases claudin proteins between enterocytes, increasing intestinal permeability and allowing the translocation of dietary antigens, metabolic endotoxins, and bacterial lipopolysaccharides (LPSs) from the gut into the bloodstream. LPS binds to TLR4 on immune cells, triggering an inflammatory cascade mediated by IL-6 and TNF-α secretion, which further increases the intestinal permeability [144]. In enteroendocrine cells, deficient LPS binding to TLR4 reduces GLP-1 and cholecystokinin secretion [145], highlighting the dual role of TLR4 in metabolic homeostasis [127].The alteration of the molecular composition of extracellular vesicles (EV). EV are lipid-bound nanostructures released by donor microbial cells and internalized by target cells, enabling the transfer of bioactive molecules such as nucleic acids, proteins, lipids, and metabolites. An altered EV composition has been linked to metabolic dysfunction, impaired immune cell recruitment, dysregulated adipocyte signaling and thermogenesis, and macrophage and T cell dysfunction in AT [151]. Recent studies describe how microbial and adipocyte-derived EV transport bioactive proteins, metabolites, cytokines, and lipid mediators, thereby regulating Th1/Th17 cell differentiation and suppressing Treg induction in VAT, a paradigm expanding the concept of the gut–adipose immunometabolic communication [153,154].

Taken together, these findings have led to the proposal of strategies combining gut microbiota modulation through the following:–Weight loss via BS to restore microbial balance and thereby normalize SCFAs and EV profiles, as well as immune cell function [155].–Capsaicin supplementation to reduce inflammation via the downregulation of NF-κβ, NLRP3, and LPS activation [147].–Application of novel multi-omics frameworks (metagenomics, metabolomics, and proteomics) and mass spectrometry platforms integrated with artificial intelligence to measure the production and function of bioactive proteins, inflammatory mediators, enzymes, SCFAs, and EV from fecal samples and/or AT [149,150]. These could serve as novel molecular targets and/or diagnostic biomarkers, enabling personalized intervention strategies targeting the gut–immune interface [153], thereby opening the door to an unprecedented analytical resolution in obesity research [154] (Figure 6).

### 5.4. Regenerative Approaches with AT and Adipose-Derived Stem Cell Transplantation for Metabolic and Immune Modulation

AT is enriched in mesenchymal stem/stromal cells, which are relatively easy to harvest and isolate and are predominantly located in BAT. These cells secrete a broad spectrum of cytokines, growth factors, and nucleic acids (e.g., miRNAs). Their secretome not only stimulates resident stem cells within tissues but also modulates immune cell activity, thereby constituting a regenerative and immunomodulatory platform in metabolic disorders such as T2D. EV derived from ADSCs have been shown to attenuate MIOR, enhance insulin sensitivity, and promote tissue repair [156]. The transplantation of BAT (a type of AT essential for energy expenditure) in place of WAT, from healthy normal-weight donors to obese recipients, may reduce glucose levels and IR. This effect is attributed to improved hepatic AKT signaling and the action of fibroblast growth factor 21 (FGF21), a hormone that lowers blood glucose and lipid levels by activating AT metabolism [157,158,159], which suggests potential therapeutic benefits in cardiometabolic diseases [157,158,160]. Notably, obesity has been shown to compromise the reparative capacity and functional properties of both autologous and transplanted ADSCs, highlighting the imperative to standardize bioengineering methodologies for their isolation, expansion, and preconditioning [161].

**Figure 6 ijms-26-09867-f006:**
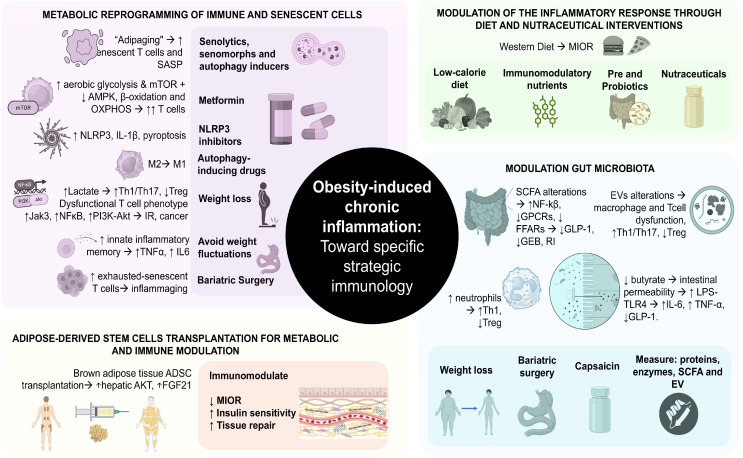
Obesity-induced chronic inflammation toward specific strategic immunology. Abbreviations: ↑: increase; ↓: decrease, →: leads to This figure was designed according to [143,144,145,146,147,148,149,150,151,154,156,157,158,159,160,162,163,164].

## 6. Conclusions

Obesity has been identified as a factor that disrupts the delicate balance of immune cells within AT, driving it toward a pro-inflammatory state. This shift partially explains the systemic effects of chronic inflammation. Overall, it can be concluded that, as obesity progresses, both the number and pro-inflammatory activity of immune cells increases. Conversely, when obesity is reduced through BS, particularly with mixed techniques, the number and activity of these cells also decrease. However, some inconsistencies in leukocyte population levels have been reported, both in obesity and after BS, possibly due to [144,158] methodological heterogeneity, biological individuality (set of unique biological characteristics of each human individual that can generate epistemological problems related to validation, decision-making, and clinical conclusions), or differences in the type and size of the study populations.

The pathophysiological and molecular mechanisms by which AT expansion in individuals with obesity and in those undergoing BS leads to complex and dynamic changes in the immune cell number and function remain incompletely elucidated. A major, detailed investigation into the understanding and organization of each process involved in this inflammatory response is still lacking. Nevertheless, evaluating the immunometabolic profile through a lymphocyte subpopulation analysis in patients with obesity may provide more comprehensive and objective clinical insights.

On the other hand, studies have investigated what may be possible through the modulation of the inflammatory response through diet and nutraceutical interventions; the metabolic reprogramming of immune and senescent cells via pharmacological interventions, weight loss, and BS; the immunomodulation of the GM; and regenerative approaches with AT and ADSC transplantation for metabolic and immune modulation. Although the above points still need further investigation, they could enable the development of more specialized and precise preventive, diagnostic, and therapeutic strategies, thus helping to prevent obesity-related complications that impair quality of life and increase mortality in the affected population.

## Figures and Tables

**Figure 1 ijms-26-09867-f001:**
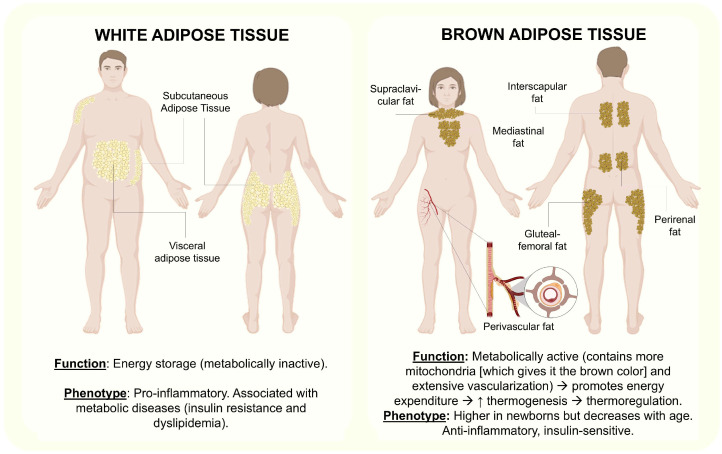
Distribution and function of AT. Shows the phenotypic differences between WAT and BAT. Abbreviations: ↑: increase; →: leads to. This figure was designed according to [2,3,4].

**Figure 2 ijms-26-09867-f002:**
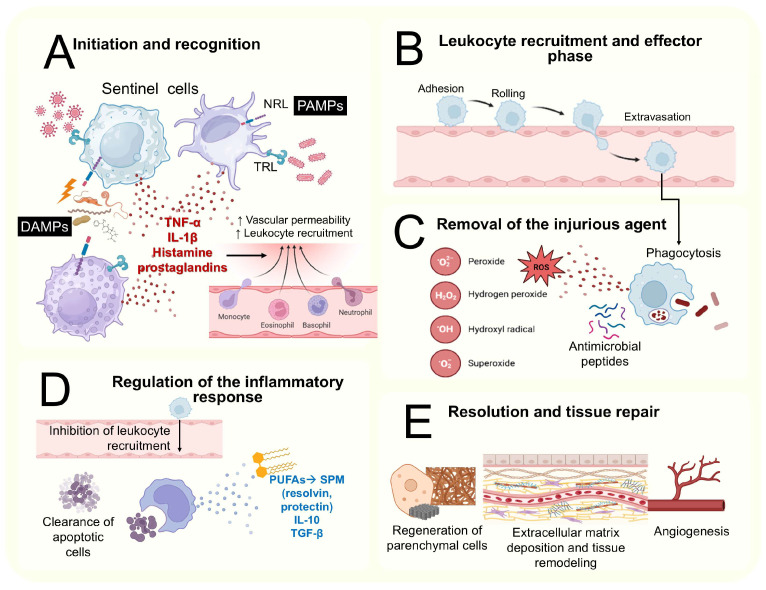
The sequential phases of the inflammatory response. This figure summarizes the basis of the inflammatory process. (**A**) Initiation and recognition. (**B**) Leukocyte recruitment and effector phase. (**C**) Removal of the injurious agent. (**D**) Regulation of the inflammatory response. (**E**) Resolution and tissue repair. Abbreviations: ↑: increase; →: leads to. This figure was designed according to [22,23,25,26].

**Figure 3 ijms-26-09867-f003:**
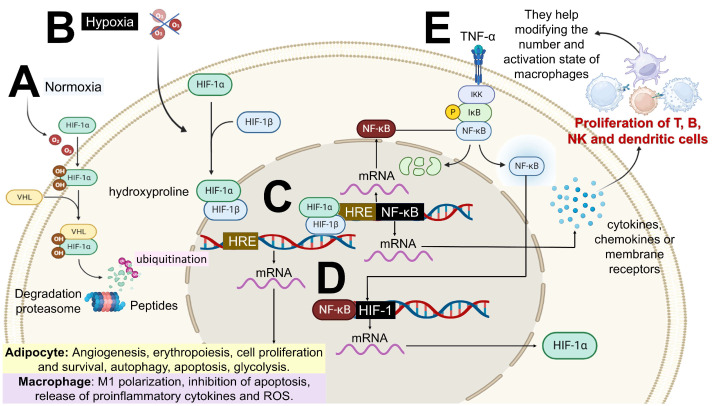
Molecular mechanisms of hypoxia and its relationship with inflammation. (**A**) Hypoxia-inducible factor 1 alpha (HIF-1α) degradation; (**B**) HIF-1α activation by hypoxia; (**C**) NF-κβ, NF-κβ activation by hypoxia; (**D**) HIF-1α regulation by hypoxia; and (**E**) TNF-α secretion and immune cell activation. This figure was designed according to [33,34].

**Figure 4 ijms-26-09867-f004:**
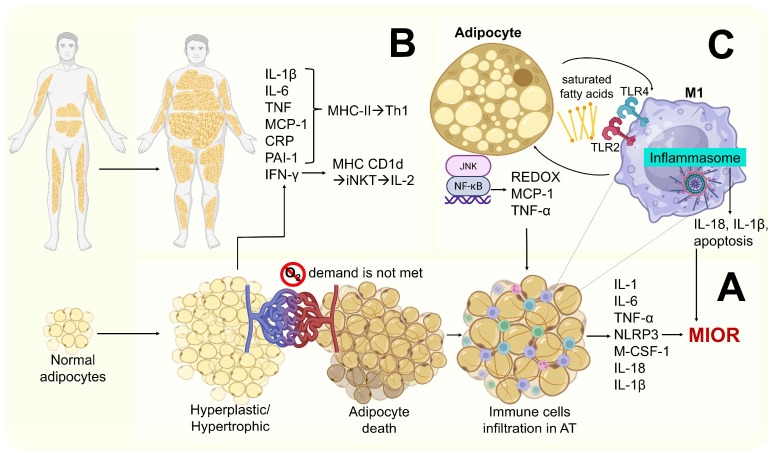
Mechanisms of obesity-induced inflammation. (**A**) Systemic distribution of adipocytes; (**B**) pathophysiology of adipocyte hypertrophy and hyperplasia and inflammation-mediating molecules. (**C**) Lipoperoxidation and MIOR. This figure was designed according to [15,34,35,45,47].

**Figure 5 ijms-26-09867-f005:**
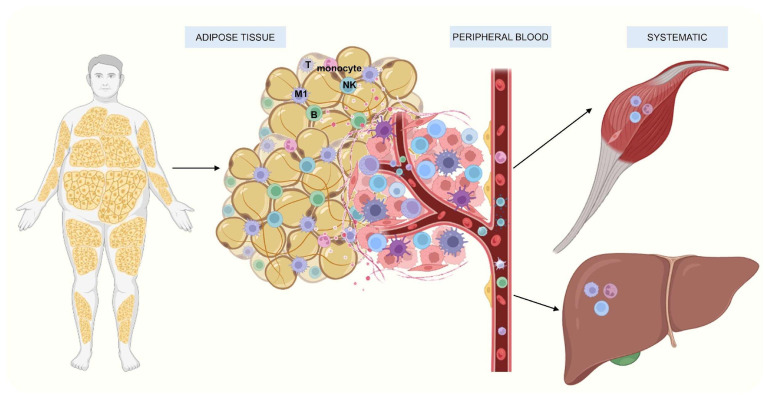
Amplification of TA hypertrophy and hyperplasia in the local and systemic immune response [32,35].

## Data Availability

In this research no new data was created.

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
