# Peer review of "Cellular Immunity in Obesity: Pathophysiological Insights and the Impact of Bariatric Surgery"

_ijms, 2025, doi:10.3390/ijms26209867_

Round 1

Reviewer 1 Report (New Reviewer)

Comments and Suggestions for Authors

Thank you for the opportunity to review your comprehensive and integrative review of the molecular and cellular mechanisms underlying obesity-induced low-grade chronic inflammation, with a particular focus on the role of cellular immunity. The authors synthesize a broad array of literature, including recent findings on immunometabolic dysfunction, bariatric surgery outcomes, and emerging therapeutic strategies such as metabolic reprogramming and gut microbiota modulation.

While the manuscript has great potential, there are some English barriers that in my opinion limit the full and relevant impact. Perhaps the authors will consider using a language editing service to address this limitation. The manuscript contains numerous grammatical errors, awkward phrasing, and inconsistent terminology. 

In revising, you may also want to consider shortening and focusing your sentences to emphasize your key points you are trying to make.

While the manuscript is through, it is in some cases redundant. I would suggest a revision of the structure to limit redundancy. There is also a lot of terms used, and some are not defined and/or are novel. A table of terms and definitions would be helpful as a reader.

It would be helpful as a reader to better integrate the figures into the overall structure of the manuscript. In doing so you could treat these as critical/important to your stated goal.

You have cited your own work significantly, which is fine as long as you emphasize that in the introduction and mention it as a potential limitation. If you wanted to be more objective, you could include other citations.

Comments on the Quality of English Language

The manuscript presents valuable insights and is scientifically sound, but it requires substantial language editing and structural refinement (to reduce length and redundancy).

Author Response

REVIEWER 1

Comments and Suggestions for Authors

Thank you for the opportunity to review your comprehensive and integrative review of the molecular and cellular mechanisms underlying obesity-induced low-grade chronic inflammation, with a particular focus on the role of cellular immunity. The authors synthesize a broad array of literature, including recent findings on immunometabolic dysfunction, bariatric surgery outcomes, and emerging therapeutic strategies such as metabolic reprogramming and gut microbiota modulation.

Answer: Thank you for your valuable comments.

While the manuscript has great potential, there are some English barriers that in my opinion limit the full and relevant impact. Perhaps the authors will consider using a language editing service to address this limitation. The manuscript contains numerous grammatical errors, awkward phrasing, and inconsistent terminology. 

Answer: Thanks for your feedback! Grammatical errors, awkward phrasing, and inconsistent terminology have been corrected.

In revising, you may also want to consider shortening and focusing your sentences to emphasize your key points you are trying to make.

Answer: We have tried to shorten and focus the sentences to emphasize our      key points throughout the document.

While the manuscript is through, it is in some cases redundant. I would suggest a revision of the structure to limit redundancy. There is also a lot of terms used, and some are not defined and/or are novel. A table of terms and definitions would be helpful as a reader.

Answer: We have corrected redundancies and defined all terms in the text (when they are first mentioned) and in abbreviations' table.

It would be helpful as a reader to better integrate the figures into the overall structure of the manuscript. In doing so you could treat these as critical/important to your stated goal.

Answer: We have integrated the figures with the text. Likewise, we have created a general figure that summarizes the article. Take a look at the graphical abstract

You have cited your own work significantly, which is fine as long as you emphasize that in the introduction and mention it as a potential limitation. If you wanted to be more objective, you could include other citations.

Answer: We reduced the number of times our article was cited. Thanks for the feedback.

Comments on the Quality of English Language

The manuscript presents valuable insights and is scientifically sound, but it requires substantial language editing and structural refinement (to reduce length and redundancy).

Answer: Thank you, we have submitted our manuscript to a review and refinement of English grammatical structure.

Reviewer 2 Report (New Reviewer)

Comments and Suggestions for Authors

This is a very timely issue because obesity is a significant global health problem. Since 1975, its prevalence has more than tripled worldwide. According to data from the Centers for Disease Control and Prevention (CDC) in the United States, 40.3% of adults were obese between 2021 and 2023.

Title and abstract

There is a discrepancy between the title and the objective stated in the abstract. The title suggests that the article will address the pathophysiology of obesity-induced chronic inflammation. However, the objective only mentions lymphocyte subpopulations and summarizes the anti-inflammatory effects of current therapeutic procedures.

I suggest changing the title, and I would appreciate it if you could do so.

Introduction

The introduction is suitable for experts familiar with the subject and for readers eager to learn.

However, I would like to emphasize that acute inflammation is a physiological process because it leads to full tissue regeneration and the restoration of homeostasis. This process is disrupted in obesity. Additionally, tissue regeneration and the resolution of inflammation are accompanied by a "lipid mediator shift," influenced by the systemic lipid profile.

Please add a section discussing the shift from pro-inflammatory eicosanoids to pro-resolving mediators.

2.1. Cellular immunity

This paragraph contains both too much and too little information. While it is a good idea for the authors to illustrate the complexity of the immune system, the wording is poor. The chapter (pages 118–134) needs to be rewritten.

2.2. Basis of the inflammatory process

The paragraph is correct but incomplete. Due to the length of the manuscript, it is unlikely that the chapter will be expanded. Therefore, I suggest adding a figure illustrating the cellular and soluble mediators of inflammation and their interactions.

2.3. Mechanism of low-grade inflammation associated with obesity

Well-written, accurate description.

2.4. Cancer and low-grade chronic inflammation associated with obesity

I would divide the section into two parts:

  1. Low-grade chronic inflammation associated with obesity.
  2. The consequences and complications of chronic inflammation. I would expand the section on "low-grade chronic inflammation associated with obesity" because it is incomplete.

3. Cellular immunity and its participation in chronic inflammation associated with obesity

4. Changes in cellular immunity in individuals with obesity

Sections 3 and 4 should be merged. Currently, the emphasis is shifted towards lymphocytes (mainly T cells) among cellular elements. More information is needed regarding the role of myeloid cells.

5. Changes in cellular immunity in obese individuals undergoing bariatric surgery

Section 5 is well-written and accurate. However, I would only include it in the manuscript if the title were changed.

Author Response

REVIEWER 2

Comments and Suggestions for Authors

This is a very timely issue because obesity is a significant global health problem. Since 1975, its prevalence has more than tripled worldwide. According to data from the Centers for Disease Control and Prevention (CDC) in the United States, 40.3% of adults were obese between 2021 and 2023.

Answer: Thank you for your motivational comments.

Title and abstract

There is a discrepancy between the title and the objective stated in the abstract. The title suggests that the article will address the pathophysiology of obesity-induced chronic inflammation. However, the objective only mentions lymphocyte subpopulations and summarizes the anti-inflammatory effects of current therapeutic procedures.

I suggest changing the title, and I would appreciate it if you could do so.

Answer: We have changed the title to "Cellular Immunity in Obesity: Pathophysiological Insights and the Impact of Bariatric Surgery" to achieve consistency with the purpose of the article and the abstract. Thanks.

Introduction

The introduction is suitable for experts familiar with the subject and for readers eager to learn.

However, I would like to emphasize that acute inflammation is a physiological process because it leads to full tissue regeneration and the restoration of homeostasis. This process is disrupted in obesity. Additionally, tissue regeneration and the resolution of inflammation are accompanied by a "lipid mediator shift," influenced by the systemic lipid profile.

Please add a section discussing the shift from pro-inflammatory eicosanoids to pro-resolving mediators.

Answer: Look at lines 94-97, we have mentioned that acute inflammation, unlike chronic inflammation (in obesity), reestablishes homeostasis through inflammation-resolving processes.

And at the lines 185-194      where we talk about the difference between acute and chronic inflammation, we delve a little deeper into the resolution of inflammation accompanied by a "lipid mediator shift," influenced by the systemic lipid profile and discussing the shift from pro-inflammatory eicosanoids to pro-resolving mediators.

2.1. Cellular immunity

This paragraph contains both too much and too little information. While it is a good idea for the authors to illustrate the complexity of the immune system, the wording is poor. The chapter (pages 118–134) needs to be rewritten.

Answer: Text was modified please see lines 148-159

2.2. Basis of the inflammatory process

The paragraph is correct but incomplete. Due to the length of the manuscript, it is unlikely that the chapter will be expanded. Therefore, I suggest adding a figure illustrating the cellular and soluble mediators of inflammation and their interactions.

Answer: We have created a new figure that completes section 2.2. Basis of the inflammatory process. Take a look at figure 2.

2.3. Mechanism of low-grade inflammation associated with obesity

Well-written, accurate description.

Answer: Thank you for your words of motivation

2.4. Cancer and low-grade chronic inflammation associated with obesity

I would divide the section into two parts:

  1. Low-grade chronic inflammation associated with obesity.
  2. The consequences and complications of chronic inflammation. I would expand the section on "low-grade chronic inflammation associated with obesity" because it is incomplete.

Answer: 2.4 section was divided into the two suggested sections. And the session of consequences was expanded. Look at lines 389-424.

  1. Changes in cellular immunity in individuals with obesity

Sections 3 and 4 should be merged. Currently, the emphasis is shifted towards lymphocytes (mainly T cells) among cellular elements. More information is needed regarding the role of myeloid cells.

Answer: Sections 3 and 4 were merged and in this section, emphasis was placed on lymphoid cells. Thanks for the suggestion.

  1. Changes in cellular immunity in obese individuals undergoing bariatric surgery

Section 5 is well-written and accurate. However, I would only include it in the manuscript if the title were changed.

Answer: We have changed the title to "Cellular Immunity in Obesity: Pathophysiological Insights and the Impact of Bariatric Surgery" to achieve consistency with the purpose of the article, the abstract and in the section where bariatric surgery is discussed. Thanks.

Reviewer 3 Report (New Reviewer)

Comments and Suggestions for Authors

Dear Editors of the International Journal of Molecular Sciences and Authors of the manuscript ijms-3848173, I appreciate the invitation sent to me to assess this manuscript’s suitability for publication in this critical journal. Obesity triggers low-grade chronic inflammation by activating immune cells, especially in visceral adipose tissue. As fat cells enlarge, they recruit macrophages that release inflammatory cytokines like TNF-α and IL-6, which contribute to systemic inflammation. This disrupts normal immune function, promoting insulin resistance and metabolic disturbances. Additionally, increased fatty acids can further activate immune responses, leading to a cycle of inflammation that exacerbates obesity-related diseases such as type 2 diabetes, cardiovascular disease, and certain cancers. Targeting this inflammatory process could offer new treatments for these conditions. I believe that this manuscript has true potential to be published in this vital journal. However, there are some concerns that I would like to address with the Authors before the Editor decides. Since the manuscript already delves into most of the necessary mechanisms underlying their review scope, I suggest major revisions.

MAJOR REVISIONS

  1. In the "Introduction" section, please clearly state the purpose of the review/study. It would be helpful to specify whether this manuscript aims to provide a comprehensive review of existing literature on the relationship between obesity, inflammation, and associated diseases. Additionally, if the focus is on exploring new therapeutic avenues, make sure to clearly articulate the goal of proposing novel treatment strategies targeting the inflammatory processes in obesity. This will provide a clear direction for readers and set expectations for the manuscript's scope.
  2. Consider including a figure in the "Introduction" to visually convey the rationale behind conducting this manuscript. The figure could illustrate the mechanisms of obesity-induced low-grade chronic inflammation and its relationship with cellular immunity. It could show the expansion of adipocytes in visceral adipose tissue and the subsequent recruitment of immune cells, particularly macrophages. The figure could also highlight how these immune cells release pro-inflammatory cytokines, such as TNF-α and IL-6, and how these inflammatory signals ultimately contribute to insulin resistance and other metabolic dysfunctions. Such a visual would provide readers with a clear understanding of the inflammatory cascade and emphasize the importance of exploring these mechanisms in the development of potential therapeutic interventions.
  3. Consider expanding the discussion on endothelial dysfunction by delving deeper into how inflammation impairs endothelial function and its direct contribution to cardiovascular disease. This could be presented in a dedicated subsection, as endothelial dysfunction is a critical and often overlooked consequence of obesity-related inflammation. By isolating this topic, you would be able to provide a more focused and detailed exploration of how pro-inflammatory cytokines, like TNF-α and IL-6, affect the endothelium and contribute to processes such as vascular stiffness, impaired nitric oxide production, and increased blood clotting. This is especially important since endothelial dysfunction is a precursor to atherosclerosis and other cardiovascular diseases, making it a crucial link between obesity, inflammation, and cardiovascular risk. A dedicated subsection would allow for a more comprehensive discussion and emphasize the need for targeted interventions aimed at improving endothelial health in obese individuals.
  4. Consider including a discussion in the subsection of “New Perspectives” on the transplantation of adipose tissue and adipose-derived stem cells as a tool for studying metabolic physiology and for potential therapeutic applications. This approach could offer a novel perspective on how adipose tissue transplantation might not only provide insights into metabolic regulation but also serve as a strategy for treating obesity-related diseases. Exploring the therapeutic potential of adipose-derived stem cells could add depth to the manuscript by highlighting emerging regenerative strategies aimed at reducing inflammation and improving metabolic and immune function. This could also open new avenues for treating obesity, metabolic syndrome, and cardiovascular diseases.
  5. Consider expanding on “Alternative and Complementary Therapies”, particularly focusing on nutraceutical interventions. Compounds like curcumin, resveratrol, and berberine have shown promise in modulating inflammatory pathways and improving metabolic health. Discussing how these compounds can complement traditional pharmacological treatments could offer a more holistic approach to managing obesity and its related immune dysfunction, linked by low-grade inflammation. This perspective could provide valuable insights into integrative medicine and the potential for combining conventional and natural therapies to address chronic inflammation, immune, and metabolic dysfunctions.
  6. It would be valuable to include a figure that separates fat tissue types by their phenotype, distribution, and function. Specifically, you could illustrate how different fat depots—such as visceral versus subcutaneous fat versus brown—differ in terms of insulin resistance and inflammatory profiles. Visceral fat, for example, is often more insulin-resistant and inflammatory compared to subcutaneous fat. This would help clarify how these phenotypic differences in fat tissue function contribute to obesity-related comorbidities. By breaking down fat tissues in this way, you can provide a clearer understanding of the pathophysiological variations that exist across individuals, depending on their fat distribution and metabolic profiles (phenotypic interactions).
  7. There are many paragraphs, such as Lines 117-134 on Page 3, that have one or very few references. In the entire manuscript, please try to include more references in cases like this specific one, with no more than three sentences without a reference.

MINOR REVISIONS

  1. Please revise the manuscript to avoid the overuse of words like "surprisingly" (e.g., Line 311, Page 8). Such terms can diminish the scientific tone of the manuscript. Instead, consider using more neutral language that directly presents the findings without implying subjectivity or surprise. This will help maintain a more formal and objective presentation of the research.

I want to thank you so much for your collaboration and consideration. I'm looking forward to receiving a revised version of this manuscript soon.

With best regards and always available,

The Reviewer.

Author Response

REVIEWER 3

Comments and Suggestions for Authors

Dear Editors of the International Journal of Molecular Sciences and Authors of the manuscript ijms-3848173, I appreciate the invitation sent to me to assess this manuscript’s suitability for publication in this critical journal. Obesity triggers low-grade chronic inflammation by activating immune cells, especially in visceral adipose tissue. As fat cells enlarge, they recruit macrophages that release inflammatory cytokines like TNF-α and IL-6, which contribute to systemic inflammation. This disrupts normal immune function, promoting insulin resistance and metabolic disturbances. Additionally, increased fatty acids can further activate immune responses, leading to a cycle of inflammation that exacerbates obesity-related diseases such as type 2 diabetes, cardiovascular disease, and certain cancers. Targeting this inflammatory process could offer new treatments for these conditions. I believe that this manuscript has true potential to be published in this vital journal. However, there are some concerns that I would like to address with the Authors before the Editor decides. Since the manuscript already delves into most of the necessary mechanisms underlying their review scope, I suggest major revisions.

MAJOR REVISIONS

  1. In the "Introduction" section, please clearly state the purpose of the review/study. It would be helpful to specify whether this manuscript aims to provide a comprehensive review of existing literature on the relationship between obesity, inflammation, and associated diseases. Additionally, if the focus is on exploring new therapeutic avenues, make sure to clearly articulate the goal of proposing novel treatment strategies targeting the inflammatory processes in obesity. This will provide a clear direction for readers and set expectations for the manuscript's scope.

Answer: We have changed the title to "Cellular Immunity in Obesity: Pathophysiological Insights and the Impact of Bariatric Surgery" to achieve consistency with the objective of the article. Look at lines 49-54, 116-126 where we clarify and integrate the objective and all sections of this review.

  1. Consider including a figure in the "Introduction" to visually convey the rationale behind conducting this manuscript. The figure could illustrate the mechanisms of obesity-induced low-grade chronic inflammation and its relationship with cellular immunity. It could show the expansion of adipocytes in visceral adipose tissue and the subsequent recruitment of immune cells, particularly macrophages. The figure could also highlight how these immune cells release pro-inflammatory cytokines, such as TNF-α and IL-6, and how these inflammatory signals ultimately contribute to insulin resistance and other metabolic dysfunctions. Such a visual would provide readers with a clear understanding of the inflammatory cascade and emphasize the importance of exploring these mechanisms in the development of potential therapeutic interventions.

Answer: We have created a general figure that summarizes the article and what we object. Take a look at the graphical abstract.

  1. Consider expanding the discussion on endothelial dysfunction by delving deeper into how inflammation impairs endothelial function and its direct contribution to cardiovascular disease. This could be presented in a dedicated subsection, as endothelial dysfunction is a critical and often overlooked consequence of obesity-related inflammation. By isolating this topic, you would be able to provide a more focused and detailed exploration of how pro-inflammatory cytokines, like TNF-α and IL-6, affect the endothelium and contribute to processes such as vascular stiffness, impaired nitric oxide production, and increased blood clotting. This is especially important since endothelial dysfunction is a precursor to atherosclerosis and other cardiovascular diseases, making it a crucial link between obesity, inflammation, and cardiovascular risk. A dedicated subsection would allow for a more comprehensive discussion and emphasize the need for targeted interventions aimed at improving endothelial health in obese individuals.

Answer: Look at lines 541-568 here's an in-depth subsection on "endothelial dysfunction" as you suggested. Thanks

Consider including a discussion in the subsection of “New Perspectives” on the transplantation of adipose tissue and adipose-derived stem cells as a tool for studying metabolic physiology and for potential therapeutic applications. This approach could offer a novel perspective on how adipose tissue transplantation might not only provide insights into metabolic regulation but also serve as a strategy for treating obesity-related diseases. Exploring the therapeutic potential of adipose-derived stem cells could add depth to the manuscript by highlighting emerging regenerative strategies aimed at reducing inflammation and improving metabolic and immune function. This could also open new avenues for treating obesity, metabolic syndrome, and cardiovascular diseases.

Answer: Take a look at lines 865-886 lines and figure 6, subtopic 5.4 is included. Regenerative approaches with AT and adipose-derived stem cells transplantation for metabolism and immune modulation.

  1. Consider expanding on “Alternative and Complementary Therapies”, particularly focusing on nutraceutical interventions. Compounds like curcumin, resveratrol, and berberine have shown promise in modulating inflammatory pathways and improving metabolic health. Discussing how these compounds can complement traditional pharmacological treatments could offer a more holistic approach to managing obesity and its related immune dysfunction, linked by low-grade inflammation. This perspective could provide valuable insights into integrative medicine and the potential for combining conventional and natural therapies to address chronic inflammation, immune, and metabolic dysfunctions.

Answer: Look at lines 751-755 and figure 6, we have modified the subvention 5.1. "Modulation of the inflammatory response through diet and nutraceutical interventions" where      nutraceutical interventions (curcumin, resveratrol, and berberine) are described.

  1. It would be valuable to include a figure that separates fat tissue types by their phenotypedistribution, and function. Specifically, you could illustrate how different fat depots—such as visceral versus subcutaneous fat versus brown—differ in terms of insulin resistance and inflammatory profiles. Visceral fat, for example, is often more insulin-resistant and inflammatory compared to subcutaneous fat. This would help clarify how these phenotypic differences in fat tissue function contribute to obesity-related comorbidities. By breaking down fat tissues in this way, you can provide a clearer understanding of the pathophysiological variations that exist across individuals, depending on their fat distribution and metabolic profiles (phenotypic interactions).

Answer: Take a look at figure 1. Phenotype, distribution, and function of adipose tissue. Thank you for your suggestion.

  1. There are many paragraphs, such as Lines 117-134 on Page 3, that have one or very few references. In the entire manuscript, please try to include more references in cases like this specific one, with no more than three sentences without a reference.

Answer: We have tried to add more references for the same idea.

MINOR REVISIONS

  1. Please revise the manuscript to avoid the overuse of words like "surprisingly" (e.g., Line 311, Page 8). Such terms can diminish the scientific tone of the manuscript. Instead, consider using more neutral language that directly presents the findings without implying subjectivity or surprise. This will help maintain a more formal and objective presentation of the research.

I want to thank you so much for your collaboration and consideration. I'm looking forward to receiving a revised version of this manuscript soon.

With best regards and always available,

The Reviewer.

Answer: We have improved the scientific tone in our writing by trying to use neutral language without insinuating subjectivity or surprise. Thank you.

We look forward to your prompt response.

Kind regards.

Round 2

Reviewer 1 Report (New Reviewer)

Comments and Suggestions for Authors

Technical edits look great. You have incorporated the figures into the manuscript better and the graphical abstract is a nice addition. You just need to address some more English language concerns.

Comments on the Quality of English Language

While the English writing in this version is greatly improved there are still some inconsistencies that need to be addressed. Below are some examples, but the entire manuscript needs a comprehensive review.

  1. "Cellular immune dysfunction in obesity es not restricted," I think the 
    "es" should be "is"
  2. Consistent use of capital letters. For example you inconsistently capitalize "obesity" vs. "obesity.
  3. Inconsistent use of hyphens in compound adjectives (e.g., “low-grade inflammation” vs. “low\-grade inflammation”)
  4. Several sentences are overly long and as a result confusing. There are several sentences with respect to HIF-1 like this.
  5. You use the terms "bio-individuality" and "trained innate-inflammatory memory" but you don't really define them and explain them differently throughout the manuscript. If these are critical then you need to be more consistent with their use.

Author Response

REVIEWER 1, Roud 2

Comments and Suggestions for Authors

Technical edits look great. You have incorporated the figures into the manuscript better and the graphical abstract is a nice addition. You just need to address some more English language concerns.

Comments on the Quality of English Language

1- While the English writing in this version is greatly improved there are still some inconsistencies that need to be addressed. Below are some examples, but the entire manuscript needs a comprehensive review.

Answer: Thank you, we have again thoroughly reviewed the writing and grammar of this manuscript using an English proofreading service.

2- "Cellular immune dysfunction in obesity es not restricted," I think the

"es" should be "is"

Answer: Take a look at line 130. We've corrected the error you highlighted. Thank you.

3- Consistent use of capital letters. For example you inconsistently capitalize "obesity" vs. "obesity.

Inconsistent use of hyphens in compound adjectives (e.g., “low-grade inflammation” vs. “low\-grade inflammation”)

Answer: Take a look lines 124 we have eliminated the capital letter where it does not correspond.

And, Take a look lines 34, 52, 97, 100, 184, 606, 654, 701, 719. We have unified the term "low-grade chronic inflammation" according to the medical subject headings (MeSH): Cifuentes M, Verdejo HE, Castro PF, et al. Low-Grade Chronic Inflammation: a Shared Mechanism for Chronic Diseases. Physiology. 2025;40(1):0. doi:10.1152/physiol.00021.2024

4- Several sentences are overly long and as a result confusing. There are several sentences with respect to HIF-1 like this.

Answer: We have corrected long and confusing hours in the manuscript. And take a look at section 2.3. we have eliminated repetitive ideas. Thanks

5- You use the terms "bio-individuality" and "trained innate-inflammatory memory" but you don't really define them and explain them differently throughout the manuscript. If these are critical then you need to be more consistent with their use.

Answer: Take a look lines 781-783. We have corrected and defined the term "biological Individuality". And take a look lines 695-701. We have corrected, defined and contextualized the term “trained immunity”

We look forward to your prompt response.

Kind regards.

Atte: The Authors

Reviewer 2 Report (New Reviewer)

Comments and Suggestions for Authors

I am satisfied with the answers to the questions and comments. I accept the corrections to the manuscript. In its current form, I believe it will be a valuable summary for the scientific community.

Author Response

REVIEWER 2. Roud 2

Comments and Suggestions for Authors

I am satisfied with the answers to the questions and comments. I accept the corrections to the manuscript. In its current form, I believe it will be a valuable summary for the scientific community.

Answer: Thanks for your valuable intervention in the revision of our manuscript.

Reviewer 3 Report (New Reviewer)

Comments and Suggestions for Authors

Authors, thank you for your cooperation throughout this process. I appreciate your willingness to provide the revised manuscript. After reviewing it, I find it acceptable for publication. Congratulations!

Author Response

REVIEWER 3, Roud 2

Comments and Suggestions for Authors

Authors, thank you for your cooperation throughout this process. I appreciate your willingness to provide the revised manuscript. After reviewing it, I find it acceptable for publication. Congratulations!

Answer: Thanks for your valuable intervention in the revision of our manuscript.

We look forward to your prompt response.

Kind regards.

Atte: The Authors

This manuscript is a resubmission of an earlier submission. The following is a list of the peer review reports and author responses from that submission.

Round 1

Reviewer 1 Report

Comments and Suggestions for Authors

The manuscript "Mechanisms of Obesity-Induced Low-Grade Chronic Inflammation and Its Relationship with Cellular Immunity" generally aims to establish the characteristics of the immune response in individuals with obesity , focused on identifying immune characteristics in low-grade chronic inflammation.

The article is interesting, with  an original objective. Indeed, many studies have demonstrated the participation of cells and molecules of the immune response, suggesting that obesity favors an inflammatory state, it  twill be interesting if authors indicated a definition of low grade inflammation, which could be more comprehensive to identify the rôle of cella as macrophages, lymphocytes in a chronic inflammatory process.

v.gr chap 1 «  These alterations activate both innate and adaptive immune cells and stimulate the production of low-molecular-weight signaling molecules known as cy tokines messenger proteins and glycoproteins that orchestrate immune responses, mediate intercellular communication, and regulate both systemic and local inflammation »

Chap 2 : Leuckocytes are the primary cells responsible for mediating immune responses. Paragraph L94-109  and L 112 The term “inflammation” is derived from the Latin word inflammatio, meaning “to  ignite” or “to set on fire”, and is often identified by the suffix -itis, which denotes a non- specific bodily response to harmful stimuli. Inflammation initially presents as an acute process but may persist and become chronic.

 Inclusion of chaper 5 should better sustained. The presentation of Chapter 5, which involves changes in cellular immunity in obese individuals undergoing bariatric surgery, is not indicated as an alternative in the abstract, and the results shown should be better justified within the scope of the work; otherwise, they do not support the objective of the work. Are there other alternatives? It would be an alternative, and there is only a comparison against analyses of individuals who did not undergo bariatric surgery or some variants. Is an alternative for obese treatment? Or for chronic low inflammation ?

Minor comments

L194 "high mobility group box 1 (HMGV1) protein." The abbreviation should be HMGB1.

L205 "Why are the descriptions of other mechanisms associated with the immunometabolic obesity response (IMOR) presented in smaller letters than the rest of the text?" They are equally important.

L265 "produces more cytosines" like IL-6, TNF α, IFN-γ, TGF-β, MCP-1, and IL-1β...must we say cytokines?"

The  figures are very nice and interesting; unfortunately, the footnotes are very extensive. It is suggested that they should be more specific; information on the mechanisms and results of activation/inhibition should be included in the text.

Author Response

The manuscript "Mechanisms of Obesity-Induced Low-Grade Chronic Inflammation and Its Relationship with Cellular Immunity" generally aims to establish the characteristics of the immune response in individuals with obesity, focused on identifying immune characteristics in low-grade chronic inflammation. The article is interesting, with an original objective. Indeed, many studies have demonstrated the participation of cells and molecules of the immune response, suggesting that obesity favors an inflammatory state,

  • Answer: Thank you very much for your kind comment

It will be interesting if authors indicated a definition of low grade inflammation, which could be more comprehensive to identify the role of cell as macrophages, lymphocytes in a chronic inflammatory process.

v.gr chap 1 « These alterations activate both innate and adaptive immune cells and stimulate the production of low-molecular-weight signaling molecules known as cytokines messenger proteins and glycoproteins that orchestrate immune responses, mediate intercellular communication, and regulate both systemic and local inflammation »

  • Answer: Take a look line 85 to 102; ; 105-116, and 140-187. We have described the definition of low-grade inflammation associated with obesity with the premises it suggests.

Chap2 : Leuckocytes are the primary cells responsible for mediating immune responses. Paragraph L94-109 and L 112 The term “inflammation” is derived from the Latin word inflammatio, meaning “to  ignite” or “to set on fire”, and is often identified by the suffix -itis, which denotes a non- specific bodily response to harmful stimuli. Inflammation initially presents as an acute process but may persist and become chronic.

  • Answer: Take a look line 81 to 85. The paragraph was placed in another section, placing first the description of the inflammatory process and then the inflammation associated with obesity.

Inclusion of chaper 5 should better sustained. The presentation of Chapter 5, which involves changes in cellular immunity in obese individuals undergoing bariatric surgery, is not indicated as an alternative in the abstract, and the results shown should be better justified within the scope of the work; otherwise, they do not support the objective of the work. Are there other alternatives? It would be an alternative, and there is only a comparison against analyses of individuals who did not undergo bariatric surgery or some variants. Is an alternative for obese treatment? Or for chronic low inflammation?

  • Answer: Take a look at lines 49-51 and 577-583. We've added an introduction to chapter 5 to coherently link what you explained. We've also added to the summary objective where we suggest that bariatric surgery may be part of the treatment to reduce obesity-associated inflammation. Thank you.

Minor comments

L194 "high mobility group box 1 (HMGV1) protein." The abbreviation should be HMGB1.

  • Answer: Take a look chapter 2.3 at line 196. We've corrected that abbreviation. Thank you.

L205 "Why are the descriptions of other mechanisms associated with the immunometabolic obesity response (IMOR) presented in smaller letters than the rest of the text?" They are equally important.

  • Answer: Take a look chapter 2.3 at lines 217 to 246. We've relocated descriptions of other mechanisms associated with low-grade inflammation, which include metainflammation (MIOR), to the main text (not the figure caption). Thank you.

L265 "produces more cytosines" like IL-6, TNF α, IFN-γ, TGF-β, MCP-1, and IL-1β...must we say cytokines?"

  • Answer: Take a look chapter 3.1 at line 431 and 432. We've corrected the spelling. Thank you for your detailed feedback.

The figures are very nice and interesting; unfortunately, the footnotes are very extensive. It is suggested that they should be more specific; information on the mechanisms and results of activation/inhibition should be included in the text.

  • Answer: Take a look lines 212 to 215 of figure caption 1. Lines 217-246 and figure caption 2 lines 318-323. To figure caption 3 please see lines 511-513. We have reduced the length of the figure caption and have specified information about the mechanisms and results of activation/inhibition in the text.

Reviewer 2 Report

Comments and Suggestions for Authors

The review entitled " Mechanisms of Obesity-Induced Low-Grade Chronic Inflammation and its Relationship with Cellular Immunity" by Rivera-Carranza T et al., describes links between obesity and chronic inflammation as also cellular immunity. It describes at length modes leading to chronic inflammation and components involved in cellular immunity.
The main problem of the present review is a major question that comes to mind: What it adds on top of so many reviews appearing on the topic frequently. Reading in the review-article reminds examining of a textbook.   It covers the subject in an "old school" tedious manner, with no relation to the latest development in the field.
Specifically, a few relevant comments:
-    It is well established that obesity correlates with increased cancer incidence, and the mechanism underlying this relationship is the subject of intense research. Therefore, the link made here to "low grade inflammation", which is an initial stage of cancer - is not appropriate.    

-    Current studies on the impact of obesity on tumor-infiltrating CD8 + T cell response in cancer is not addressed adequately. This is of major current interest in the field. 

-    How obesity affects immune checkpoint blockade (ICB) therapy in the combat of tumor growth? This subject needs to be broadly discussed.

-    There is no reference to the issue of the "obesity paradox" namely, obesity as a risk factor for cancer on one hand but associated with improved treatment outcomes in cancer patients, on the other (Lennon H, Sperrin M, Badrick E, Renehan AG. The obesity paradox in cancer: a review. Curr Oncol Rep. 2016; 18(9):56. doi:10.1007/s11912-016-0539-4).

-    Importantly, outlining a list of factors that either are upregulated or inhibited under obesity is tedious and hard to follow. It is recommended instead to describe a coherent narrative in a context dependent problem.

-    Here is one example of a recently published relevant paper highlighting advancement in the field. It should be cited and reviewed accordingly. Piening A et al., Obesity-related T cell dysfunction impairs immunosurveillance and increases cancer risk. Nat Commun. 2024 Apr 2;15(1):2835.

Author Response

The review entitled " Mechanisms of Obesity-Induced Low-Grade Chronic Inflammation and its Relationship with Cellular Immunity" by Rivera-Carranza T et al., describes links between obesity and chronic inflammation as also cellular immunity. It describes at length modes leading to chronic inflammation and components involved in cellular immunity.

  • Answer: Thank you very much for your kind comment

The main problem of the present review is a major question that comes to mind: What it adds on top of so many reviews appearing on the topic frequently. Reading in the review-article reminds examining of a textbook.   It covers the subject in an "old school" tedious manner, with no relation to the latest development in the field.

  • Answer:

We sincerely appreciate your comment and the opportunity to improve our manuscript. We acknowledge that the topic of obesity-induced low-grade chronic inflammation has been addressed in previous reviews. However, we would like to emphasize that our contribution aims to stand out by offering an integrative and up-to-date perspective, with a specific focus on the relationship between cellular immunity—particularly lymphocyte subpopulations—and bariatric surgery. This connection remains underexplored in depth in the recent literature.

In addition, we have incorporated recent scientific advances from 2021 to 2025, including:

  • Novel immunometabolic functions of lymphocyte subtypes (CD4+, CD8+, CD28+, CD62+, etc.);
  • Molecular mechanisms such as NLRP3 inflammasome activation, mitochondrial dysfunction, and the role of HIF-1α in adipose tissue inflammation.
  • Quantitative and qualitative changes in immune cell profiles following bariatric procedures (LSG, LRYGB), emphasizing immunological and clinical markers that have not been comprehensively synthesized in prior reviews.

These elements are particularly detailed in the new Section 6 “Recent advances in the field of obesity-induced low-grade chronic inflammation and its relationship with cellular immunity” of the manuscript, where we highlight recent findings on epigenetic modulation, gut microbiota alterations, metabolic inflammation, and immune reprogramming. Rather than simply reiterating established knowledge, we strive to critically synthesize and contextualize emerging evidence from a cellular immunology perspective, providing a framework for novel therapeutic strategies.

We have further revised the “Conclusions” and “Recent Advances” sections to reinforce the originality of our approach and underline its potential clinical relevance, while also identifying current gaps in the field. We are grateful for your thoughtful feedback, which has helped us better articulate the distinctive contribution and added value of this review within the growing body of literature.

Specifically, a few relevant comments:
-    It is well established that obesity correlates with increased cancer incidence, and the mechanism underlying this relationship is the subject of intense research. Therefore, the link made here to "low grade inflammation", which is an initial stage of cancer - is not appropriate.  

  • Answer: Take a look line 114. We made it clear that low-grade chronic inflammation is related to obesity, which not only contributes to the initial stage of cancer but also contributes to its perpetuation. And new chapter 2.4 lines 334-397 and 718- 727 we have described how low-grade inflammation associated with obesity generates and perpetuates cancer.

Current studies on the impact of obesity on tumor-infiltrating CD8 + T cell response in cancer is not addressed adequately. This is of major current interest in the field. 

  • Answer: Take a look new chapter 2.4, lines 369 to 376. We have mentioned the impact of obesity and weight loss on tumor-infiltrating CD8 + T cell response in cancer.

How obesity affects immune checkpoint blockade (ICB) therapy in the combat of tumor growth? This subject needs to be broadly discussed.

  • Answer: Take a look in lines 349 to 376, we have discussed checkpoint blockade.

-    There is no reference to the issue of the "obesity paradox" namely, obesity as a risk factor for cancer on one hand but associated with improved treatment outcomes in cancer patients, on the other (Lennon H, Sperrin M, Badrick E, Renehan AG. The obesity paradox in cancer: a review. Curr Oncol Rep. 2016; 18(9):56. doi:10.1007/s11912-016-0539-4).

  • Answer: Take a look line 390 to 397; 702-727. We have addressed this paradox, thanks.

Importantly, outlining a list of factors that either are upregulated or inhibited under obesity is tedious and hard to follow. It is recommended instead to describe a coherent narrative in a context dependent problem.

  • Answer: the context of the factors upper regulated or inhibited was expanded please see chapter 4 and 5.

Here is one example of a recently published relevant paper highlighting advancement in the field. It should be cited and reviewed accordingly. Piening A et al., Obesity-related T cell dysfunction impairs immunosurveillance and increases cancer risk. Nat Commun. 2024 Apr 2;15(1):2835.

  • Answer: A new section abording the advances in the field was included, please see lines 685 to 727. Thank you.
